# Relationship of the Nigrostriatal Tract with the Motor Function and the Corticospinal Tract in Chronic Hemiparetic Stroke Patients: A Diffusion Tensor Imaging Study

**DOI:** 10.3390/healthcare10040731

**Published:** 2022-04-14

**Authors:** Sung Ho Jang, Min Jye Cho

**Affiliations:** Department of Physical Medicine and Rehabilitation, College of Medicine, Yeungnam University, Namku, Taegu 42415, Korea; strokerehab@hanmail.net

**Keywords:** nigrostriatal tract, motor function, corticospinal tract, diffusion tensor tractography, diffusion tensor imaging

## Abstract

This study investigated the relationship of the nigrostriatal tract (NST) with motor function and the corticospinal tract (CST) using diffusion tensor tractography in chronic hemiparetic stroke patients. Forty-three consecutive patients with putaminal hemorrhage in the chronic stage were recruited. The Motricity Index was used to evaluate the motor function of affected hemiparetic extremities. The fractional anisotropy and the tract volume of ipsilesional NST and ipsilesional CST were acquired. The tract volume (*Rho* = 0.824) of ipsilesional NST and fractional anisotropy (*r* = 0.682) and the tract volume (*Rho* = 0.886) of ipsilesional CST showed a strong positive correlation with the Motricity Index score. The fractional anisotropy of ipsilesional NST showed moderate positive correlations with the fractional anisotropy (*r* = 0.449) and tract volume (*Rho* = 0.353) of ipsilesional CST. The tract volume of ipsilesional NST showed strong positive correlations with the fractional anisotropy (*Rho* = 0.716) and the tract volume (*Rho* = 0.799) of ipsilesional CST. The regression model showed that the tract volumes of ipsilesional NST and ipsilesional CST were positively associated with the Motricity Index score (Adjusted R^2^ = 0.763, F = 45.998). Mediation analysis showed that the tract volume of ipsilesional CST partially mediated the effects of the tract volume of ipsilesional NST on the Motricity Index score (*z* = 3.34). A close relationship was found between ipsilesional NST and the motor function of affected extremities in chronic hemiparetic patients with putaminal hemorrhage. Moreover, ipsilesional NST influenced the motor function of affected extremities indirectly through ipsilesional CST.

## 1. Introduction

Stroke is one of the most common causes of disability in adults, and more than 50% of stroke patients have been reported to experience motor weakness [1]. Several neural tracts are involved in the motor function in the human brain [2,3,4]. The elucidation of the relationship of each neural tract with the motor function is clinically important for rehabilitating stroke patients. Among the neural tracts for the motor function, the relationships of the corticospinal tract (CST) and corticoreticulospinal tract with motor weakness in stroke patients are well-known [2,3,4,5,6,7,8,9,10]. By contrast, little is known about the other neural tracts, including the nigrostriatal tract (NST).

The NST, one of the major dopaminergic pathways, is a neural pathway that originates from the substantia nigra pars compacta in the midbrain, terminates to the dorsal striatum (the caudate nucleus and putamen), and affects voluntary movement through the basal ganglia motor loops [11]. The NST facilitates the initiation and execution of voluntary movement through the direct pathways, where the dopamine D1 receptor is projected, whereas it also inhibits unwanted muscle contractions through indirect pathways where the dopamine D2 receptor is expressed via the basal ganglia [12,13,14]. Several studies have reported the relationship between the reduced activity of the nigrostriatal system and the severity of motor weakness in patients with neurological diseases, such as Parkinson’s disease, multiple system atrophy, and Huntington’s disease, using brain mapping methods including positron emission tomography and tensor-based morphometry [15,16,17]. On the other hand, the relationship of the NST with the motor function in stroke patients has not been reported. In addition, the techniques used in the above studies were limited in a precise estimation of the NST.

The CST, which originates mainly from the primary motor cortex of the motor cortical areas receiving basal ganglia output, is the most important neural tract for fine control of the amount and velocity of voluntary movement of the distal extremity [18,19,20]. Previous animal studies reported that activation of the dopamine D2 receptor expressed by neurons through the indirect pathway that passes through the striatum could modulate the neuronal activity of the primary motor cortex by increasing the neuronal spike firing rate [21,22,23,24,25,26]. These results suggest that the CST, which originates mainly from the primary motor cortex, can intervene in the effect of the NST on motor function [21,22,23,24,25,26]. Therefore, studies on the relationship between the NST and CST in stroke patients are clinically important for identifying the pathophysiology of motor weakness and developing rehabilitative strategies.

The development of diffusion tensor imaging (DTI), which generates images based on estimations of the diffusion of water molecules in microstructures, has enabled an evaluation of the entire microstructural feature of the white matter in the human brain. In particular, diffusion tensor tractography (DTT) derived from DTI enables a three-dimensional reconstruction and visualization of the NST and CST [27,28,29]. Many DTT-based studies have reported the relationship between motor function and the CST state in stroke patients [2,3,5,6,7,8,10]. On the other hand, no study on the relationship of the NST with the motor function and the CST has been reported. The current study tested the following hypotheses: First, the NST state affects the motor function of hemiparetic stroke patients. Second, the NST and CST are associated with the motor function in hemiparetic stroke patients.

In this study, by using DTT, we investigated the relationship of the NST with motor function and the CST in chronic hemiparetic stroke patients.

## 2. Materials and Methods

### 2.1. Subjects

Forty-three consecutive patients with putaminal hemorrhage (27 men, 16 women; mean age 50.60 ± 10.76 years; range, 24–68 years) were enrolled. The inclusion criteria for this study were as follows: (1) first-ever stroke, (2) age: 20–69 years, (3) spontaneous putaminal hemorrhage confirmed by a neuroradiologist, (3) hemiparesis of the affected extremities contralateral to putaminal hemorrhage lesion, (4) DTI scanning and motor function evaluation were performed at the chronic stage (more than three months after onset), and (5) no previous history of psychiatric and neurological disease. All patients underwent comprehensive rehabilitative therapy, including movement therapies provided by physical and occupational therapists (motor strengthening of the affected upper and lower extremities, and exercises for trunk stability and control, static and dynamic balance training on sitting and standing positions; twice/day, 40 min/time, five days/week), and neuro-muscular electrical stimulation for the affected finger extensors and ankle dorsiflexors (twice/day, 20 min/time, five days/week). Informed consent was obtained from all the subjects. This study was performed retrospectively in accordance with the requirements of the Declaration of Helsinki research guidelines, and the institutional review board of a Yeungnam university hospital approved the study protocol (IRB number: YUMC 14 March 2021).

### 2.2. Motor Function Evaluation

The Motricity Index was used to assess the motor function of the affected extremities in the chronic stage of putaminal hemorrhage (9.18 ± 8.72 months after putaminal hemorrhage onset). The Motricity Index was scored as follows: for all movements except for grip, 0; no movement, 28; palpable contraction in muscle, but no movement, 43; visible movement without gravity, but not full range, 57; full range of movement against gravity, 76; full range of movement against resistance, but weaker than normal, 100; normal, for grip, 0; no movement, 34; beginning of prehension, 58; able to grip a cube, without gravity, 67; able to grip and hold a cube, against gravity, 79; grips against a weak pull but weaker than the other side, 100; normal [30]. The total Motricity Index score is the average of the Motricity Index scores for shoulder abductor, elbow flexor, prehension, hip flexor, knee extensor, and ankle dorsiflexor [30]. The reliability and validity of the Motricity Index are well-established [31].

### 2.3. Diffusion Tensor Imaging and Tractography

The DTI data were obtained during motor function evaluation using a sensitivity-encoding head coil on a 1.5 T Philips Gyroscan Intera (Hoffman-LaRoche Ltd., Best, The Netherlands) scanner with single-shot echo-planar imaging and navigator echo. Sixty-seven contiguous slices (acquisition matrix = 96 × 96; reconstruction matrix = 192 × 192; field of view = 240 mm × 240 mm; TR = 10,726 ms; TE = 76 ms, b = 1000 s/mm^2^, NEX = 1, and thickness = 2.5 mm) were acquired for each of the 32 noncollinear diffusion-sensitizing gradients. Eddy current image distortion and head motion effects were adjusted using affine multi-scale two-dimensional registration from the Oxford Centre for Functional Magnetic Resonance Imaging of Brain (FMRIB) Software Library (FSL: www.fmrib.ox.ac.uk/fsl) [32]. Evaluations of the NST and the CST were performed using DTI Studio software (CMRM, Johns Hopkins Medical Institute, Baltimore, MD, USA), which was based on the fiber assignment continuous tracking (FACT) algorithm. The NST was reconstructed by placing the first regions of interest manually on the substantia nigra at the midbrain on the fractional anisotropy map and locating the second regions of interest on the striatum on the fractional anisotropy map [33,34]. The CST was tracked by assigning the first and second regions of interest manually on the anterior blue portion of the upper pons and lower pons on the axial image of the color map, respectively [35]. Fiber tracking of the NST was started at the center of a seed voxel with a fractional anisotropy of >0.1 and ended at a voxel with a fractional anisotropy of >0.1 and a tract turning angle of <70° with an option of cut operation on the axial images (Figure 1) [29,32]. On the other hand, fiber tracking of the CST was started at the center of the voxel with a fractional anisotropy of >0.2 and ended at the voxel with a fractional anisotropy of <0.2 and a tract turning angle of < 60° [36,37].

### 2.4. Statistical Analysis

Statistical analysis was conducted using SPSS 21.0 for Windows (SPSS, Chicago, IL, USA). The Shapiro–Wilk test was used to evaluate the normality of the DTT (fractional anisotropy and tract volume), the parameters of the ipsilesional NST, motor function (Motricity Index score), and the DTT parameters of the ipsilesional CST. The tract volumes of the ipsilesional NST and the ipsilesional CST did not satisfy normality (*p* < 0.05). According to the normality of the data, Pearson and Spearman correlation analyses were used to evaluate the significance of the correlations of the DTT parameters of the ipsilesional NST with the motor function (Motricity Index score) and DTT parameters of the ipsilesional CST. The correlation coefficient represents the strength (≥0.1 and <0.3, weak correlation; ≥0.3 and <0.5; moderate correlation; ≥0.5, strong correlation) and direction (positive or negative) of the relationship between the two variables [38]. After correlation analyses, the multiple linear regression analysis was conducted to evaluate the DTT parameters as independent variables that significantly affect the motor function (Motricity Index score) as a dependent variable.

Mediation analysis was used to evaluate whether the tract volume of the ipsilesional CST mediates the relationship between the tract volume of the ipsilesional NST and motor function (Motricity Index score). Referring to the Baron and Kenny mediational model, the tract volume of the ipsilesional CST plays a mediating role in the following conditions: (1) In the first step, the independent variable (tract volume of the ipsilesional NST) significantly affects the mediator variable (tract volume of the ipsilesional CST). (2) In the second step, the independent variable (tract volume of the ipsilesional NST) significantly affects the dependent variable (Motricity Index score). (3) In the third step, the independent variable (tract volume of the ipsilesional NST) and mediator variable (tract volume of the ipsilesional CST) significantly affect the dependent variable (Motricity Index score) [39]. In the third step, full mediation exists when the independent variable has no significant effect on the dependent variable. By contrast, partial mediation exists when the standardized β of the independent variable in the third step is less than that in the second step [39]. The Sobel test was used to confirm the significance of the mediation effect; *p* values < 0.05 were considered significant [39,40].

## 3. Results

Table 1 lists the demographic and clinical data of the patients. According to the correlation analyses, the fractional anisotropy value of the ipsilesional NST showed no significant correlation with the Motricity Index score (*p* > 0.05). In contrast, the tract volume of the ipsilesional NST showed a strong positive correlation with the Motricity Index score (*Rho* = 0.824; *p* < 0.05). The fractional anisotropy value and tract volume of the ipsilesional CST revealed strong positive correlations with the Motricity Index score (fractional anisotropy value *r* = 0.682; tract volume *Rho* = 0.886; *p* < 0.05). The fractional anisotropy value of the ipsilesional NST showed moderate positive correlations with the fractional anisotropy value and tract volume of the ipsilesional CST (fractional anisotropy value *r* = 0.449; tract volume *Rho* = 0.353; *p* < 0.05). In addition, the tract volume of the ipsilesional NST showed strong positive correlations with the fractional anisotropy value and tract volume of the ipsilesional CST (fractional anisotropy value *Rho* = 0.716; tract volume *Rho* = 0.799; *p* < 0.05). Table 2 lists the results of multiple linear regression analysis of DTT parameters of the ipsilesional NST and the ipsilesional CST for motor function (Motricity Index score). The tract volumes of the ipsilesional NST and the ipsilesional CST were positively associated with the Motricity Index score (Adjusted R^2^ = 0.763, F = 45.998, *p* < 0.05). On the other hand, the fractional anisotropy value of the ipsilesional CST was not related to the Motricity Index score (*p* > 0.05).

Table 3 presents the mediation analysis results of the role of the tract volume of the ipsilesional CST as a mediator of the relationship between the tract volume of the ipsilesional NST and motor function (Motricity Index score). In the case of the Motricity Index score as a dependent variable, in the first step, the tract volume of the ipsilesional NST had a significant association with the tract volume of the ipsilesional CST (*t* = 10.977, *p* < 0.05). In the second step, the tract volume of the ipsilesional NST showed a significant association with the Motricity Index score (*t* = 9.881, *p* < 0.05). In the third step, the tract volumes of the ipsilesional NST (*t* = 2.674) and ipsilesional CST (*t* = 3.344) had significant associations with the Motricity Index score (*p* < 0.05), and the standardized β of the tract volume of the ipsilesional NST was lower in the third step (β = 0.403) than that in the second step (β = 0.839). In summary, the tract volume of the ipsilesional CST partially mediated the relationship between the tract volume of the ipsilesional NST and the Motricity Index score (*z* = 3.34, *p* < 0.05).

## 4. Discussion

This study examined the relationship of the ipsilesional NST with the motor function and the ipsilesional CST in chronic hemiparetic patients with putaminal hemorrhage using DTT. The results are summarized as follows. First, the tract volume of the ipsilesional NST and the fractional anisotropy value and tract volume of the ipsilesional CST showed strong positive correlations with the Motricity Index score. Second, the fractional anisotropy value (moderate) and tract volume (strong) of the ipsilesional NST showed positive correlations with the fractional anisotropy value and tract volume of the ipsilesional CST. Third, the tract volumes of the ipsilesional NST and the ipsilesional CST were predictors for describing the variability of the Motricity Index score. Fourth, the tract volume of the ipsilesional CST partially mediated the effects of the tract volume of the ipsilesional NST on the Motricity Index score.

Regarding the DTT parameters, the fractional anisotropy value indicated the degree of directionality of water diffusion and the integrity of white matter microstructures, such as axons, myelin, and microtubules. Therefore, it reflects the fiber density, axonal diameter, and white matter myelination [41,42]. The tract volume represents the volume of the neural tract, indicating the number of neural fibers [42,43]. These results show that the tract volume of the ipsilesional NST and the fractional anisotropy value and tract volume of the ipsilesional CST showed strong positive correlations with the Motricity Index score, meaning that the fiber amount of the ipsilesional NST and the integrity and fiber amount of the ipsilesional CST were closely related to the motor function of patients with putaminal hemorrhage.

The fractional anisotropy value (moderate) and tract volume (strong) of the ipsilesional NST showed positive correlations with the fractional anisotropy value and tract volume of the ipsilesional CST, indicating that the integrity and fiber amount of the ipsilesional NST are closely related to the integrity and fiber amount of the ipsilesional CST. Considering the correlation coefficient, the fiber amount of the ipsilesional NST is more closely associated with the integrity and fiber amount of the ipsilesional CST than the integrity of the ipsilesional NST. These results suggest similar vulnerabilities of the NST and CST in patients with putaminal hemorrhage. These similar vulnerabilities were attributed to the close location of the NST and CST in the subcortical white matter (Figure 1).

According to multiple linear analyses of the DTT parameters of the ipsilesional NST and the ipsilesional CST for the motor function, the tract volumes of the ipsilesional NST and the ipsilesional CST were predictors for describing the variability of the Motricity Index score. The tract volumes of the ipsilesional NST and the ipsilesional CST were positively associated with the Motricity Index score in the regression model. These results show that the fiber amounts of the ipsilesional NST and ipsilesional CST influence the motor function of the affected extremities in patients with putaminal hemorrhage.

Regarding the mediation analysis, the result showing that the tract volume of the ipsilesional CST partially mediated the relationship between the tract volume of the ipsilesional NST and Motricity Index score shows that the fiber amount of the ipsilesional NST influences the motor function of the affected extremities independently or through the fiber amount of the ipsilesional CST. This result suggests that rehabilitation strategies increasing the fiber amounts of the ipsilesional NST and the ipsilesional CST appear to be needed to facilitate the recovery of the motor function of the affected extremities in patients with putaminal hemorrhage.

The CST state is closely related to the motor function of patients with brain injury [2,3,5,6,7,8,10]. On the other hand, little is known about the NST. Several studies have shown that the nigrostriatal system is related to motor function in patients with neurological diseases [15,16,17]. In 1990, Brooks et al. reported that the reduced nigrostriatal dopaminergic system in the putamen was correlated with the severity and duration of locomotor disability in patients with Parkinson’s disease and multiple system atrophy using positron emission tomography [15]. In 2005, Kipps et al. reported that the significant loss of gray matter volume limited to the substantia nigra pars compacta and striatum was associated with the severity of motor weakness in patients with Huntington’s disease using tensor-based morphometry [16]. Subsequently, in 2014, Obeso et al., using positron emission tomography, reported that the dopamine deficit from the substantia nigra pars compacta to the striatum was associated with the reducing ability of the initiation and execution of both the automatic and voluntary movements in a Parkinson’s disease patient [17]. To the best of the authors’ knowledge, this is the first study to demonstrate the relationship of the NST with the motor function and the CST in patients with putaminal hemorrhage.

Some limitations of this study should be considered. First, DTT analysis is operator-dependent and can induce false-positive and false-negative results due primarily to crossing fibers or the partial volume effect [44]. Second, because this study was conducted retrospectively, only the Motricity Index score was used to evaluate the motor function, and we could not include other evaluation scales for motor function including the functional independence measure scale. Third, the patients in this study could have more severe clinical features than those in a more general population of chronic patients with putaminal hemorrhage because the patients who visited the rehabilitation department of a university hospital were recruited in this study. Therefore, further prospective studies, including various motor function evaluations and considering the generality of the clinical features, should be encouraged.

## 5. Conclusions

In conclusion, a close relationship was found between the ipsilesional NST and motor function of the affected extremities in chronic hemiparetic patients with putaminal hemorrhage. In addition, the ipsilesional NST influenced the motor function of the affected extremities indirectly through the ipsilesional CST. These results suggest that both states of the ipsilesional NST and the ipsilesional CST could be clinically important in the motor outcome in patients with putaminal hemorrhage. Consequently, a rehabilitative strategy based on the states of the ipsilesional NST and the ipsilesional CST could be helpful to increase the effects of rehabilitation for improving the motor function after putaminal hemorrhage, for example, the administration of dopaminergic agonists for nigrostriatal dopamine activation.

## Figures and Tables

**Figure 1 healthcare-10-00731-f001:**
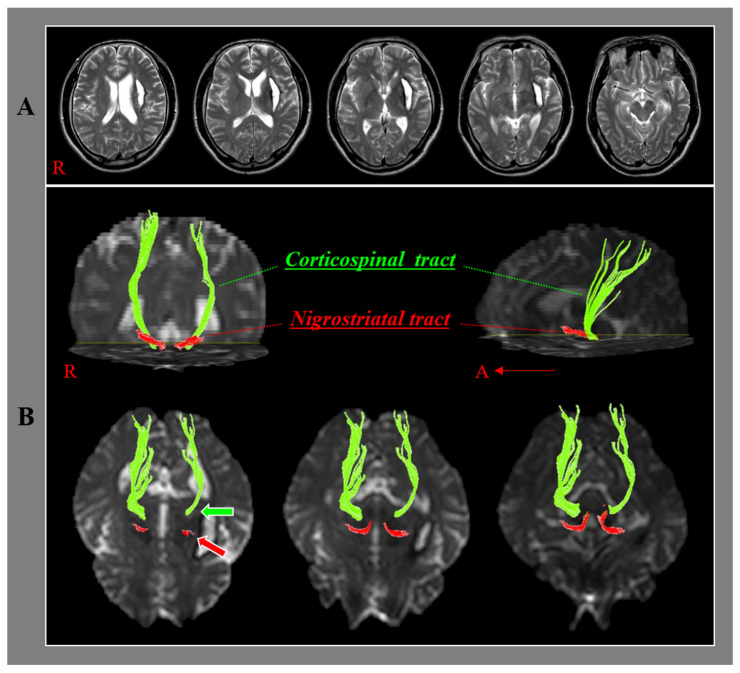
Results of diffusion tensor tractography for the nigrostriatal tract (NST) and the corticospinal tract (CST). (**A**) T2-weighted brain magnetic resonance images at the time of diffusion tensor imaging scanning in a representative patient of a putaminal hemorrhage (49-year-old male). (**B**) The ipsilesional NST and the ipsilesional CST between basal ganglia and midbrain are shown at each brain level. The ipsilesional NST (red arrow) and the ipsilesional CST (green arrow) are located in close proximity to the adjacent part of the hematoma.

**Table 1 healthcare-10-00731-t001:** Demographic and clinical data of the patients.

	Patients
Lesion side (right:left)	18:25
Mean duration to DTI (months)	9.18 ± 8.72
Motricity Index score	55.95 ± 20.92

DTI: diffusion tensor imaging; values presented are means ± standard deviations.

**Table 2 healthcare-10-00731-t002:** Multiple linear regression analysis of the diffusion tensor tractography parameters of the ipsilesional nigrostriatal tract and the ipsilesional corticospinal tract for the motor function.

Dependent Variable	Independent Variable	B	se	β	*t*	*p*	VIF
Motricity Index	Tract volume of NST	0.144	0.062	0.356	2.328	0.03 *	4.145
	Fractional anisotropy of CST	31.640	22.898	0.144	1.382	0.18	1.919
	Tract volume of CST	0.019	0.006	0.449	2.902	0.01 *	4.229
	Adjusted R^2^ = 0.763 F = 45.998 *p* = 0.00 *						

VIF: variation inflation factor, NST: nigrostriatal tract, CST: corticospinal tract, *: statistically significant at *p* < 0.05.

**Table 3 healthcare-10-00731-t003:** Mediation analysis of the role of the tract volume of the ipsilesional corticospinal tract as a mediator of the relationship between the tract volume of the ipsilesional nigrostriatal tract and motor function.

Step	Dependent Variable	Independent Variable	B	se	β	*t*	*p*	Adjusted R^2^
1	Tract volume of CST	Tract volume of NST	8.465	0.771	0.864	10.977	0.00 *	0.740
2	Motricity Index	Tract volume of NST	0.340	0.034	0.839	9.881	0.00 *	0.697
3	Motricity Index	Tract volume of NST	0.163	0.061	0.403	2.674	0.01 *	0.757
		Tract volume of CST	0.021	0.006	0.504	3.344	0.00 *	
Sobel test Z = 3.34 *								

CST: corticospinal tract, NST: nigrostriatal tract, *: statistically significant at *p* < 0.05.

## Data Availability

Data available on request due to privacy/ethical restrictions.

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
