# Peer review of "Relationship of the Nigrostriatal Tract with the Motor Function and the Corticospinal Tract in Chronic Hemiparetic Stroke Patients: A Diffusion Tensor Imaging Study"

_healthcare, 2022, doi:10.3390/healthcare10040731_

Round 1

Reviewer 1 Report

THE FIM SCALE (FUNCTIONAL INDEPENDENCE MEASURE) MUST BE USED TO MEASURE CHANGES OVER TIME IN NEUROREHABILITATION.

Author Response

Point 1) The FIM scale (functional independence measure) must be used to measure changes over time in neurorehabilitation.

Response: We totally agree with the reviewer’s comment. However, because this cross-sectional study was conducted retrospectively, we could not have the FIM scale of all subjects. We added this problem in the limitation part. We revised as follows.

  1. Discussion

Some limitations of this study should be considered. First, DTT analysis is operator-dependent and can induce false-positive and false-negative results due primarily to crossing fibers or the partial volume effect [44]. Second, because this study was conducted retrospectively, only the Motricity Index score was used to evaluate the motor function and we could not include other evaluation scales for motor function including the functional independence measure scale. Third, the patients in this study could have more severe clinical features than those in a more general population of chronic patients with putaminal hemorrhage because the patients who visited the rehabilitation department of a university hospital were recruited in this study. Therefore, further prospective studies, including various motor function evaluations and considering the generality of the clinical features, should be encouraged.

Reviewer 2 Report

The authors studied the relationship of corticospinal tract and nigrostriatal tract in patients with hemoragic stroke involving putamen. I have a few comments:

  1. Table 1 features the results therefore it should be moved to result section, and the data from table 1 should not be duplicated in the text in paragraph 2.1.
  2. The normality was checked using Kolmogorov-Smirnov test, This test is appropriate for large samples. Because the sample in this research is small, other more powerful test are better suited, e.g.  Shapiro-Wilk test.
  3. I suggest using less abbreviations (TV, FA, MI) in the text and abstract, since it is hard to read, especially for reader that will not read the whole manuscript from beginning.
  4. Where there any differences between male and female? Could duration to DTI affect the results?
  5. Were all images interpreted and analyzed by the same operator? How experienced was the operator analyzing the images?
  6. "Consequently, a rehabilitative strategy based on the states of the ipsilesional NST and the ipsilesional CST would be helpful to increase the effects of rehabilitation for improving the motor function after PH." Please change the "would" to "could" since there are not enough studies yet to suggest this.

Author Response

Point 1) Table 1 features the results therefore it should be moved to result section, and the data from table 1 should not be duplicated in the text in paragraph 2.1.

Response: We totally agree with the reviewer’s comment. We moved table 1 to the result section, and deleted the overlapped data in the text. We revised as follows.

  1. Materials and Methods

2.1. Subjects

Forty-three consecutive patients with putaminal hemorrhage (27 men, 16 women; mean age 50.60 ± 10.76 years; range, 24~68 years) were enrolled. The inclusion criteria for this study were as follows: (1) first-ever stroke, (2) age: 20-69 years, (3) spontaneous putaminal hemorrhage confirmed by a neuroradiologist, (3) hemiparesis of the affected extremities contralateral to putaminal hemorrhage lesion, (4) DTI scanning and motor function evaluation were performed at the chronic stage (more than three months after onset), and (5) no previous history of psychiatric and neurological disease. All patients underwent comprehensive rehabilitative therapy, including movement therapies provided by physical and occupational therapists (motor strengthening of the affected upper and lower extremities, and exercises for trunk stability and control, static and dynamic balance training on sitting and standing positions; twice/day, 40 minutes/time, five days/week), and neuro-muscular electrical stimulation for the affected finger extensors and ankle dorsiflexors (twice/day, 20 minutes/time, five days/week). Informed consent was obtained with all subjects. This study was performed retrospectively in accordance with the requirements of the Declaration of Helsinki research guidelines, and the institutional review board of a Yeungnam university hospital approved the study protocol (IRB number: YUMC 2021-03-014).

  1. Results

Table 1 lists the demographic and clinical data of the patients. According to the correlation analyses, the fractional anisotropy value of the ipsilesional NST showed no significant correlation with the Motricity Index score (p > 0.05). In contrast, the tract volume of the ipsilesional NST showed a strong positive correlation with the Motricity Index score (Rho = 0.824; p < 0.05).

Table 1. Demographic and clinical data of the patients.

Patients

Lesion side (right : left)

18 : 25

Mean duration to DTI (months)

9.18 ± 8.72

Motricity Index score

55.95 ± 20.92

DTI: diffusion tensor imaging; Values presented are means ± standard deviations.

Point 2) The normality was checked using Kolmogorov-Smirnov test, This test is appropriate for large samples. Because the sample in this research is small, other more powerful test are better suited, e.g.  Shapiro-Wilk test.

Response: Thank you for the reviewer’s comment. We revised the normality test from the Kolmogorov-Smirnov test to the Shapiro-Wilk test. The fractional anisotropy of the ipsilesional NST (p = 0.156) and the ipsilesional CST (p = 0.389), and the Motricity Index score (p = 0.051) satisfied normality (p > 0.05). The tract volumes of the ipsilesional NST and the ipsilesional CST did not satisfy normality (p < 0.05). We revised as follows.

  1. Materials and Methods

2.4. Statistical analysis

Statistical analysis was conducted using SPSS 21.0 for Windows (SPSS, Chicago, IL, USA). The Shapiro-Wilk test was used to evaluate the normality of the DTT [fractional anisotropy, tract volume] parameters of the ipsilesional NST, motor function (Motricity Index score), and DTT parameters of the ipsilesional CST. The tract volumes of the ipsilesional NST and the ipsilesional CST did not satisfy normality (p < 0.05).

Point 3) I suggest using less abbreviations (TV, FA, MI) in the text and abstract, since it is hard to read, especially for reader that will not read the whole manuscript from beginning.

Response: Thank you for the reviewer’s comment. We changed abbreviations (PH, MI, FA, ROI, TV) to full-terminologies (putaminal hemorrhage, Motricity Index, fractional anisotropy, region of interest, tract volume) throughout the whole manuscript.

Point 4) Where there any differences between male and female? Could duration to DTI affect the results?

Response: Thank you for the reviewer’s comment. However, the purpose of this study was to investigate the relationship of the NST with motor function and the CST in chronic hemiparetic stroke patients. Therefore, we did not consider sex differences in the results of the study. In addition, according to the purpose of this study, the duration of DTI was set at the chronic stage (more than three months after onset) following the inclusion criteria.

Point 5) Were all images interpreted and analyzed by the same operator? How experienced was the operator analyzing the images?

Response: Thank you for the reviewer’s comment. All images were analyzed and interpreted by the same operator (corresponding author: Cho MJ: 25 months experience for DTI analysis as a fulltime researcher).

Point 6) "Consequently, a rehabilitative strategy based on the states of the ipsilesional NST and the ipsilesional CST would be helpful to increase the effects of rehabilitation for improving the motor function after PH." Please change the "would" to "could" since there are not enough studies yet to suggest this.

Response: Thank you for the reviewer’s comment. We revised as follows.

  1. Conclusions

In conclusion, a close relationship was found between the ipsilesional NST and motor function of the affected extremities in chronic hemiparetic patients with putaminal hemorrhage. In addition, the ipsilesional NST influenced the motor function of the affected extremities indirectly through the ipsilesional CST. These results suggest that both states of the ipsilesional NST and the ipsilesional CST could be clinically important in the motor outcome in patients with putaminal hemorrhage. Consequently, a rehabilitative strategy based on the states of the ipsilesional NST and the ipsilesional CST could be helpful to increase the effects of rehabilitation for improving the motor function after putaminal hemorrhage, for example, administration of the dopaminergic agonists for nigrostriatal dopamine activation.

Reviewer 3 Report

This is an interesting study. I think that the authors should have to elaborate more on the rehabilition aspect. What are the usual rehabilitation strategies that are performed in these cases? What changes should be carried out in said strategies according to the finidngs of this study?

Author Response

Point 1) This is an interesting study. I think that the authors should have to elaborate more on the rehabilition aspect. What are the usual rehabilitation strategies that are performed in these cases? What changes should be carried out in said strategies according to the finidngs of this study?

Response: We totally agree with the reviewer’s comment. We revised as follows.

  1. Materials and Methods

2.1. Subjects

Forty-three consecutive patients with putaminal hemorrhage (27 men, 16 women; mean age 50.60 ± 10.76 years; range, 24~68 years) were enrolled. The inclusion criteria for this study were as follows: (1) first-ever stroke, (2) age: 20-69 years, (3) spontaneous putaminal hemorrhage confirmed by a neuroradiologist, (3) hemiparesis of the affected extremities contralateral to putaminal hemorrhage lesion, (4) DTI scanning and motor function evaluation were performed at the chronic stage (more than three months after onset), and (5) no previous history of psychiatric and neurological disease. All patients underwent comprehensive rehabilitative therapy, including movement therapies provided by physical and occupational therapists (motor strengthening of the affected upper and lower extremities, and exercises for trunk stability and control, static and dynamic balance training on sitting and standing positions; twice/day, 40 minutes/time, five days/week), and neuro-muscular electrical stimulation for the affected finger extensors and ankle dorsiflexors (twice/day, 20 minutes/time, five days/week). Informed consent was obtained with all subjects. This study was performed retrospectively in accordance with the requirements of the Declaration of Helsinki research guidelines, and the institutional review board of a Yeungnam university hospital approved the study protocol (IRB number: YUMC 2021-03-014).

  1. Conclusions

In conclusion, a close relationship was found between the ipsilesional NST and motor function of the affected extremities in chronic hemiparetic patients with putaminal hemorrhage. In addition, the ipsilesional NST influenced the motor function of the affected extremities indirectly through the ipsilesional CST. These results suggest that both states of the ipsilesional NST and the ipsilesional CST could be clinically important in the motor outcome in patients with putaminal hemorrhage. Consequently, a rehabilitative strategy based on the states of the ipsilesional NST and the ipsilesional CST could be helpful to increase the effects of rehabilitation for improving the motor function after putaminal hemorrhage, for example, administration of the dopaminergic agonists for nigrostriatal dopamine activation.
